# Dynamics of pore formation during laser powder bed fusion additive manufacturing

Aiden A. Martin[1], Nicholas P. Calta[1], Saad A. Khairallah[1], Jenny Wang[1], Phillip J. Depond[1], Anthony Y. Fong[2], Vivek Thampy[2], Gabe M. Guss[1], Andrew M. Kiss[2], Kevin H. Stone [2], Christopher J. Tassone[2], Johanna Nelson Weker [2], Michael F. Toney [2], Tony van Buuren[1] & Manyalibo J. Matthews[1]

Laser powder bed fusion additive manufacturing is an emerging 3D printing technique for the fabrication of advanced metal components. Widespread adoption of it and similar additive technologies is hampered by poor understanding of laser-metal interactions under such extreme thermal regimes. Here, we elucidate the mechanism of pore formation and liquid-solid interface dynamics during typical laser powder bed fusion conditions using in situ X-ray imaging and multi-physics simulations. Pores are revealed to form during changes in laser scan velocity due to the rapid formation then collapse of deep keyhole depressions in the surface which traps inert shielding gas in the solidifying metal. We develop a universal mitigation strategy which eliminates this pore formation process and improves the geometric quality of melt tracks. Our results provide insight into the physics of laser-metal interaction and demonstrate the potential for science-based approaches to improve confidence in components produced by laser powder bed fusion.

[1] Lawrence Livermore National Laboratory, Livermore, CA 94550, USA. [2] Stanford Synchrotron Radiation Lightsource, SLAC National Accelerator Laboratory, Menlo Park, CA 94025, USA. Correspondence and requests for materials should be addressed to T.B. (email: vanbuuren1@llnl.gov) or to M.J.M. (email: matthews11@llnl.gov)

Laser-based additive manufacturing (AM) approaches such as laser powder bed fusion (LPBF) hold the potential to revolutionize manufacturing of complex metal components in the aerospace, medical, and automotive industries[1]. LPBF is particularly attractive as it permits the production of otherwise impossible geometries via a layer-by-layer strategy to print components from a thin layer of metal powder spread on a solid metal substrate, using only a computer-aided design (CAD) file to guide the scanning of a high-power laser[2]. This strategy allows LPBF to avoid the geometric limitations and extensive tooling requirements present in conventional subtractive fabrication methods. Despite its significant advantages, widespread adoption of LPBF remains limited due to concerns over component quality and consistency[3]. Reports of mechanical properties for LPBF-produced components vary widely, which presents a significant challenge for designers[4–7] and certification authorities[8]. This variability in material quality arises from both the unusual thermal history and rapid solidification of material imposed by laser-induced heating[9–11] as well as defects introduced during the process[12,13]. To improve the confidence in components built by LPBF, a greater understanding of laser–metal interaction in this extreme thermal regime and its correlation with defect generation during the LPBF process is required.

A particularly ubiquitous class of defects in LPBF-produced components are keyhole pores[14], which form when excess energy is imparted by the laser to the melt pool. These pores act as stress concentrators and have a negative effect on mechanical properties[4,13]. While keyhole porosity is somewhat stochastic in nature, such pores have been observed in regular patterns within fabricated components[15]. These patterns of pores have been attributed to melt pool dynamics at the point where the laser turns off at the end of a linear scan routine and/or laser turn points in serpentine scan patterns[16]. Since these regions can occur thousands of times in a single component and typically near the edges of the component where the impact to mechanical properties is the most pronounced, an understanding of the laser–metal interaction process during these events could have major outcomes for the quality of components produced by LPBF.

Ex situ studies of the LPBF process have shown that overheating during changes in laser scan velocity, such as at laser turn points leads to increased evaporation of metal from the surface causing a deep keyhole depression to form. The keyhole depression is unstable and can collapse to trap inert shielding gas, such as argon, in pores within the substrate[17–19]. However, direct observation of the formation dynamics of such pores has proven elusive because any viable monitoring technique must probe subsurface, micron scale dynamics at time scales on the order of ten microseconds to capture process-relevant phenomena[20]. High-speed, in situ transmission X-ray imaging is an emerging technique for probing subsurface phenomena during LPBF processing[21–23]. The technique provides information complementary to the large body of literature describing LPBF process physics using high-speed, in situ optical probes[24–30] and has been applied to multiple materials and LPBF processing conditions to understand the physics of spatter[31] and melt pool dynamics in unsupported overhang regions[23]. However, critical subsurface information such as mechanisms of pore formation and melt pool geometry under typical processing conditions, which are ideal for study by in situ X-ray imaging, remain relatively unexplored.

Here, we perform in situ transmission X-ray imaging to probe laser–metal interactions during LPBF processing and elucidate the mechanisms leading to pore formation. We show experimentally, in a common titanium alloy (Ti–6Al–4V) used for critical applications in aerospace[32] and biomedical industries[33], that the formation of pores during changes in laser scan velocity

such as at laser turn points proceeds via the rapid collapse of the vapor depression at the surface and subsequent trapping of argon by liquid metal flowing into the void. Complementary multiphysics simulations provide high-definition insight into general trends in metal–laser interactions during LPBF processing and confirm the pore formation mechanism detailed in the experimental efforts. Based on these experimental and simulated observations, we devise and implement a successful mitigation strategy to prevent pore formation at laser turn points by modulating the laser power to compensate for melt pool overheating. An analytical model defining the dimensionless quantity normalized enthalpy[18,34] as a function of laser power, scan speed, and beam size applied to the mitigation strategy reveals that a near-constant vapor depression depth can be realized by varying the laser power to maintain a constant normalized enthalpy during laser scanning. The successful mitigation strategy is straightforward to implement and can be deployed on virtually any commercial machines using existing hardware. More profoundly, the results enhance the understanding of laser–metal interaction under realistic LPBF processing conditions and reveal a mechanism leading to the formation of pores which ultimately give rise to material performance degradation in fabricated components. The pore mitigation strategy revealed here reduces the probability of pore formation and holds the potential to significantly reduce defect density and therefore improve the reliability of a component fabricated by LPBF.

## Results

**Pores formed at the laser turn point**. The properties of pores formed at a laser turn point during LPBF processing of Ti–6Al–4V were determined as a function of laser power and scan speed. Laser power and scan speeds used for LPBF processing were consistent with typical build parameters[35]. Five hundred-micrometer-thick Ti–6Al–4V substrates with and without an approximately 60-μm-thick powder layer on the surface were irradiated by a 1070 nm, 50 μm diameter laser beam over a 2.5 mm long single turn scan pattern with 50 μm hatch spacing (Fig. 1). Transmission X-ray images were captured at 20 kHz while performing LPBF processing and the resulting image time series was analyzed to determine pore properties and formation kinetics. Figure 2a reveals the pore depth as a function of distance from the turn point under various processing conditions (for a full description of pore depth, cross-sectional area, and distance from turn point see Supplementary Fig. 1). The maximum depth of pores formed at the turn point increases as a function of laser power, and this trend is independent of steady-state scan speed. A key finding from this study is that pores are primarily formed within 200 μm of the turn point under all investigated processing conditions, with 87% of the pores observed during this study observed in this region, which is consistent with ex situ observations from full builds in stainless steel[36]. Furthermore, pores closer to the turn point are generally deeper in the material than pores formed farther from the turn point. Inspection of an X-ray image time series captured at each respective processing condition reveals that pores form very quickly on time scales comparable to the sampling rate of our measurement (50 μs). Therefore, we did not attempt to resolve from these data the time it takes for a pore to form, but instead treat pore formation as a discrete event and note the time (referred to hereafter as pore initiation time, $\tau_p$), when pores form relative to the laser turn time, $t_{turn} = 0$ in Fig. 2b. All pores observed in this study are formed after the laser passes the turn midpoint, and nearly all pores are formed 200–1000 μs after this midpoint regardless of the set scan speed (Fig. 2b). While nearly all pores associated with the turn point form during this time

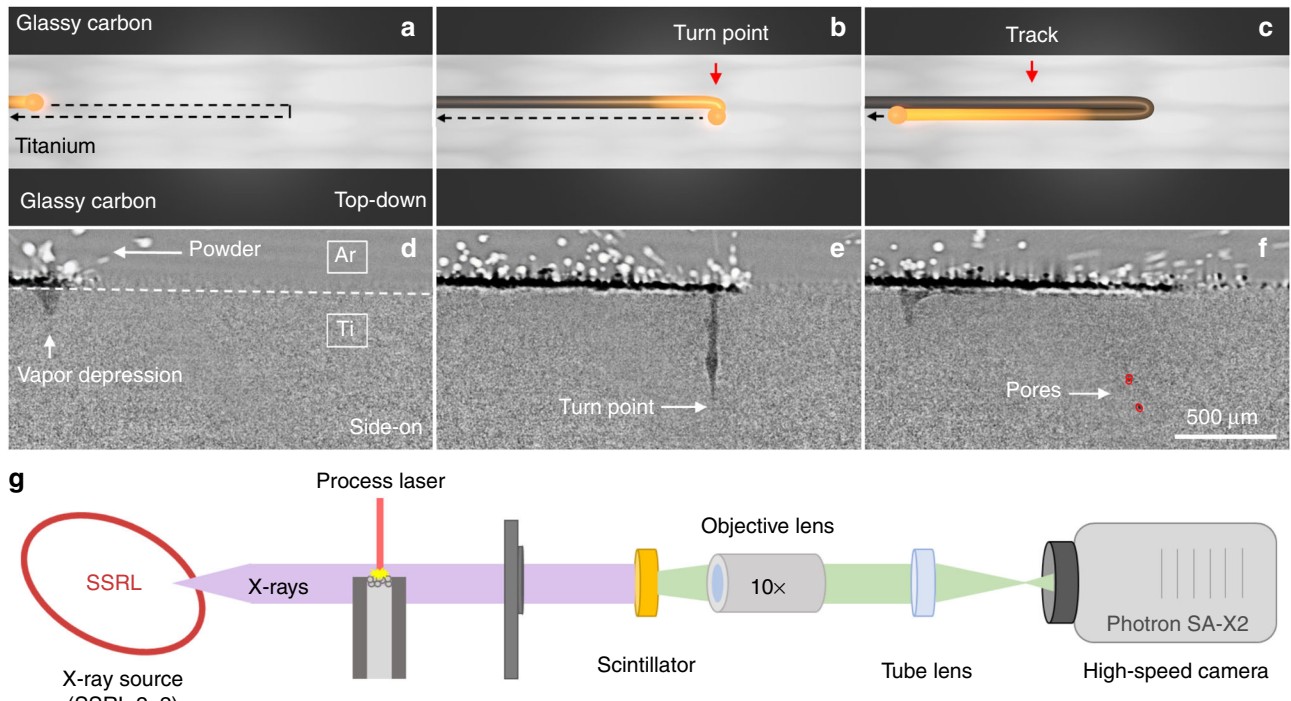

**Fig. 1** Description of a laser turn point condition and experimental configuration. **a–c** A laser turn point is defined as the condition during laser powder bed fusion (LPBF) where the laser reaches the end of a track, decelerates, shifts a prescribed hatch spacing, changes the scan direction by 180°, and then accelerates along a new track parallel and adjacent to the previous track. The black dashed line indicates laser trajectory. **d–f** Time difference ($t-t_O$), transmission X-ray images of a turn point region in Ti–6Al–4V performed at a laser power of 200 W, and scan speed of 1000 mm s$^{-1}$. **d** The laser is scanning from the left to right with spatter and powder motion above the substrate surface and a depression in the surface of the melt pool due to vapor recoil below. The titanium–argon interface is indicated by the white dashed line. **e** The laser enters the turn point region and shifts by the prescribed hatch spacing. **f** The laser is moving right to left after the turn point forming a new adjacent track and leaving behind keyhole pores. **g** Simplified schematic of the experiment configuration. A white-beam X-ray source is provided by experimental station 2-2 at the Stanford Synchrotron Radiation Lightsource (SSRL). The X-ray field of view is coincident with the 1070 nm processing laser at the Ti–6Al–4V substrate surface. Images are captured using a scintillator-based high-speed optical system

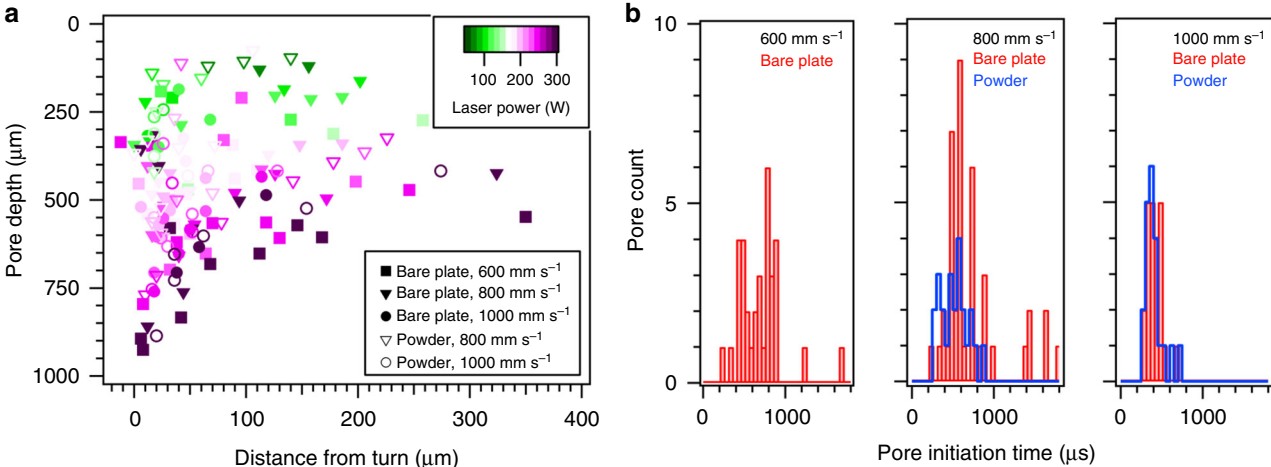

**Fig. 2** Properties of pores formed during LPBF of Ti–6Al–4V in the laser turn point region as a function of laser power and steady-state scan speed. All turn point condition scans were performed at full laser power. **a** Depth of pore relative to the substrate surface as a function of distance from the turn point of the laser. **b** Histograms of the pore initiation time, $\tau_p$, after the laser completed the turn point for three different scan speeds where $t_{turn} = 0$ μs. Each histogram includes pores produced with all laser powers (50–300 W) at the specified scan speed with (blue line) and without (red line) powder. No pores were formed in the turn point region prior to the laser turn in these experiments

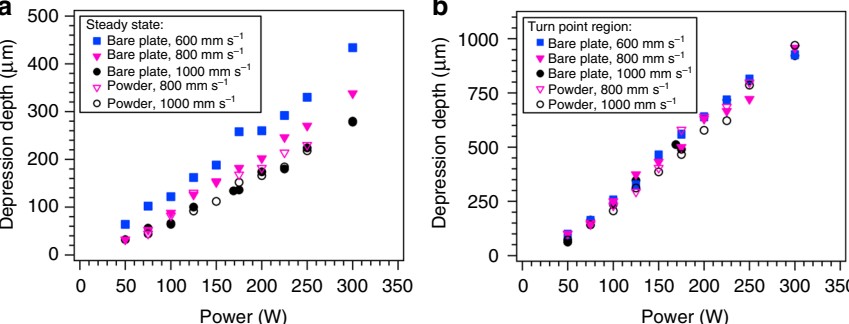

**Fig. 3** Vapor depression depth during LPBF of Ti–6Al–4V. All laser turn conditions were performed at full laser power. **a** Depression depth in the steady-state scan regime where the laser is at full scan speed as a function of laser power and scan speed. **b** Maximum depth of the vapor depression during the laser turn as a function of laser power and set point scan speed. Note the different Y-axis scale between the two panels

window, $\tau_p$ does appear to correlate with set scan speed, as pores at 1000 mm s$^{-1}$ predominantly occur (89% of the time) in less than 500 μs compared to 27% and 36% of the time at 600 and 800 mm s$^{-1}$ scan speeds, respectively. Furthermore, the results show that while careful ex situ studies can locate pores relative to scan position, only time resolved, in situ X-ray probes can with certainty, identify the time dependence of pore formation.

**Changes in surface morphology at the laser turn point.** To elucidate the mechanism of pore formation, the geometry of the melt pool surface (vapor–liquid interface) was quantitatively tracked throughout the turn point. Under these process conditions, the melt pool surface forms a depression whose shape is dominated by recoil pressure generated by metal vaporization at the melt pool surface[26]. The depth of this vapor depression during steady-state scanning and at the turn point was determined using the same processing and X-ray imaging conditions presented in Fig. 2. The vapor depression depth in Ti–6Al–4V as a function of laser power is presented in Fig. 3. Under steady-state scan conditions, the vapor depression depth increases linearly with laser power and increases with decreasing scan speed (Fig. 3a). Variations in vapor depression depth can be described in terms of changes in the localized energy density, which varies the rate of metal vaporization and therefore recoil pressure. Increasing the energy density increases the recoil pressure which drives the melt pool surface deeper into the material[17,26]. A linear scaling of vapor depression depth with laser power is not necessarily expected for keyhole mode heat transport where strong vaporization and melt pool dynamics play important roles. In this keyhole regime, the melt is rapidly displaced away from the laser beam under the effect of recoil momentum and Marangoni shear flow. The absorbed laser energy therefore not only leads to melting, but also to melt motion. The linear dependence observed here appears to indicate that the absorbed laser energy is spent mainly to melt the solid even in the keyhole regime, which helps to explain the linear scaling of melt depth behavior with power. The addition of a 60-μm powder layer on the surface did not appear to influence the dynamics of pore formation at the turn point, likely due to denudation of metal powder along the laser scan path[25]. Interestingly, for the case of the maximum vapor depression depth during the turn point, the depression relationship with power is also linear (Fig. 3b); however, the depression depth as a function of laser power is identical for all steady-state scan speeds. Inspection of the velocity of galvanometer-based X–Y scanning mirrors during the turn point indicates that at the turn point itself, the programmed steady-state scan speed has no influence on the measured scan speed (see Supplementary Fig. 2 for scanning mirror properties). Instead, the physical response time (~650 μs step response time) of the

mirrors dictates the actual scan speed during the turn point. The measured deceleration and acceleration of the mirrors approximately 500 μs pre- and post-turn varied between the scan speeds used here and approached a maximum of $1.4 \times 10^6$ mm s$^{-2}$ at a programmed steady-state value of 1000 mm s$^{-1}$. As the laser approaches the turn point, the mirrors decelerate and then accelerate back to full steady-state scan speed immediately after the turn. When the laser scan speed approaches zero at the turn point, the instantaneous energy density increases, leading to localized overheating and an increase in depression depth.

To further probe how the change in depression depth correlates with pore formation, the depression depth was determined as a function of time with a set scan speed of 1000 mm s$^{-1}$ and laser power of 100, 200, and 300 W (Fig. 4). The vapor depression depth is at steady state when it is approximately 1000 μs from the turn point. As the laser approaches 500 μs from the turn point the vapor depression depth begins to increase until it approaches a maximum at approximately 100 μs post-turn. The maximum vapor depression depth occurring post-turn is caused by a build-up of heat in the turn point region from the long dwell time of the near-stationary laser. This increase in vapor depression depth highlights a transition between the onset of keyhole mode present during steady state and a deep keyhole mode regime at the turn region. During LPBF, ideally the melt pool depth is sufficient to melt a few layers of previously processed material at the surface (up to approximately 100 μm) to fuse this material to the powder. For the case of keyhole mode welding the power density of the laser is sufficient to evaporate metal from the surface and initiate a plasma plume[18]. Evaporation of metal from the surface during keyhole mode allows the laser to drill into the material leading to the formation of a vapor depression. When the depression exceeds a depth on the order of 100 μm the deep keyhole regime is entered and a dramatic increase in the absorption of the laser power is realized due to multiple interactions between the melt pool and reflected laser[37]. This increase in laser absorption and therefore localized energy density is accompanied by formation of a high aspect ratio keyhole vapor depression as observed in Fig. 1e. After the turn, the vapor depression depth reduces to a near-steady-state regime at a time of 1000 μs post-turn. The steady-state vapor depression depth during the post-turn scan is higher than the pre-turn scan due to thermal lag associated with preheating of the material (this can be directly observed in Fig. 1a–c). Pores form at all laser powers under these processing conditions, with pores forming only post-turn during vapor depression collapse. From these direct observations, we ascribe the mechanism of pore formation at laser turn points to rapid collapse of the vapor depression. After the depression depth increases during the laser turn, the mirrors accelerate away from the turn too quickly for the

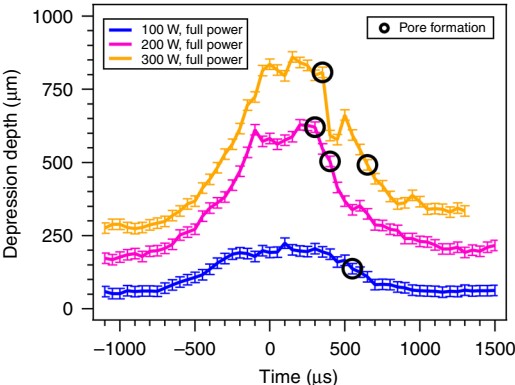

**Fig. 4** Vapor depression depth as a function of time for various laser powers at a laser turn point with a programmed steady-state scan speed of 1000 mm s$^{-1}$. These scans used a constant power, and time zero corresponds to the midpoint of the turn ($t_{\text{mid-point}} = 0 \, \mu s$). The laser is at steady-state scan speed before it begins to decelerate at approximately $-900 \, \mu s$ to initiate the turn. Minimum laser scan speed is reached at $0 \, \mu s$ after which it immediately starts to accelerate, returning to steady-state scan speed at approximately $1200 \, \mu s$. The turn midpoint was determined by analysis of the in situ X-ray images. Black circles denote pore formation events. Error bars represent uncertainty in the distance between the base of the vapor depression and the surface caused by surface roughness

depression to smoothly return to the steady state, prompting a rapid collapse of the depression. This is exemplified in Fig. 2b where the histogram of $\tau_p$ is shifted to shorter times at a steady-state scan speed of 1000 mm s$^{-1}$ compared to the 800 mm s$^{-1}$ case, which is caused by an increase in the mirror acceleration out of the turn point at the higher scan speed. The collapse of the vapor depression and subsequent pore formation can be described by hydrodynamics, and the exertion of force by gaseous metal on the molten pool surrounding the depression. Increased localized energy density in material at the turn point leads to an increase in the vapor pressure of gaseous metal above the base of the vapor depression. The increase in vapor pressure causes metal to rapidly be ejected from the vapor depression driving the depression deep into the substrate and the recoil pressure of gaseous material inside the depression overcomes the force of molten metal flow into the void. Once the scan speed returns to steady state and a decrease in temperature at the depression surface is realized the recoil pressure from evaporating metal is reduced exponentially. As the surface temperature is reduced the surface tension of the melt pool increases overcoming the force from the recoil pressure and the depression collapses[17]. Argon filled pores are then trapped in place by the quickly freezing melt pool, leaving pores trapped in the solidified material. This collapse mechanism is distinct from the traditional view of pore formation during keyhole mode, in which instabilities at the liquid metal–vapor interface stochastically form pores even in steady state[38]. Under turn point conditions where laser scan acceleration is maximum, pores are formed due to the transition of the vapor depression into a deep keyhole regime and the associated collapse of the walls of vapor depression, which is too rapid for the system to smoothly accommodate without the formation of pores.

**Multi-physics simulation**. To further our understanding of pore formation, a series of simulations were performed to ascertain the dynamics of the collapsing vapor depression. Simulations were performed using the ALE3D multi-physics software tool[39] and parameters for the validated, stainless steel (SS316L) simulation environment. The model solves the Navier–Stokes equations

coupled with the heat diffusion equation in an energy-conserving scheme, while accounting for the vapor recoil pressure and eva-porative cooling as boundary conditions using Anisimov's model[40]. Simulated laser rays strike the surface from the source in a direct line of sight and the energy deposited into the sample is determined by an effective absorption coefficient (0.25). Note that we do not employ the polarization-dependent Fresnel equations since the fiber laser source used in this study is unpolarized and thus yields a negligible absorptivity dependence on incident angle up to ~60°. The bulk of the incident laser energy is deposited over the front inclined wall of the vapor depression which consists of a flat liquid surface. The energy deposited into the front wall location dominates the melt pool response and melt pool depth via the recoil pressure. This is due to the exponential temperature dependence of the recoil physics and because the highest surface temperature is realized immediately below the laser at the point of incidence. Previously, the model was shown to predict melt pool dimensions, as well as explain the formation mechanism of other defect modes such as end-of-track pore defects[17]. Here, simulations were used to dynamically resolve the melt flow and defect formation during the turn, at high temporal (1 μs) and spatial resolution (3 μm), allowing confirmation of our experimental observations. Furthermore, the simulation enables probing of the generality of the vapor depression dynamics observed in the materials.

First, to confirm turn point dynamics in the simulated case of SS316L followed the same trend observed experimentally in Ti–6Al–4V, vapor depression and melt pool depth as a function of laser power during the laser turn point was simulated using the ALE3D multi-physics model (for further details of the simulation see Supplementary Note 1 and Supplementary Movie 1). The simulated steady-state vapor depression depth correlates with the low energy density regime (75 W, 1000 mm s$^{-1}$) measured experimentally in Ti–6Al–4V when compared using the thermal scaling laws for LPBF[34], and the transition in vapor depression depth from steady state to turn point is similar to the 50 W, 1000 mm s$^{-1}$ condition in Ti–6Al–4V (23 to 70 μm and 32 to 62 μm for the SS316 simulation and Ti6Al-4V experiment, respectively). The linear dependence of vapor depression depth with respect to the laser power was also reproduced in the SS316L simulation (see Supplementary Fig. 3a). The simulation reported a peak metal evaporative flux of approximately 1000 mol m$^{-2}$ s$^{-1}$ directly below the laser spot on the front inclined wall of the vapor depression during the turn point.

A single frame of the 3D simulation is shown in Fig. 5 for 200 W and 1500 mm s$^{-1}$ laser scanning conditions, with temperature contour lines highlighting the liquid–solid interface and color map describing the thermal gradients in the material. The steady-state-simulated vapor depression depth is ~23 μm (Fig. 5a). As the laser approaches the end of track and decelerates, the vapor depression depth increases to ~70 μm (Fig. 5b). This is a direct result of the increase in laser dwell time as the scan speed decreases for execution of the turn, and more energy is absorbed in the local volume causing the depth to increase. A small decrease in depth is observed during the simulation at the turn point due to the laser traveling across the hatch spacing of 100 μm and encountering colder material that requires additional energy to heat to melt pool temperatures. As the laser finishes the turn, a more pronounced depth increase occurs because the accumulated residual heat is higher in this pre-heated location compared to entry into the turn point. Immediately following this melt pool depth maximum, the vapor depression depth rapidly decreases as the laser accelerates away from the turn point (Fig. 5c). A significant increase of the vapor depression depth due to this overheating followed by rapid collapse (Fig. 5d) gives rise to pore formation, as liquid metal cannot fill the deep depression before

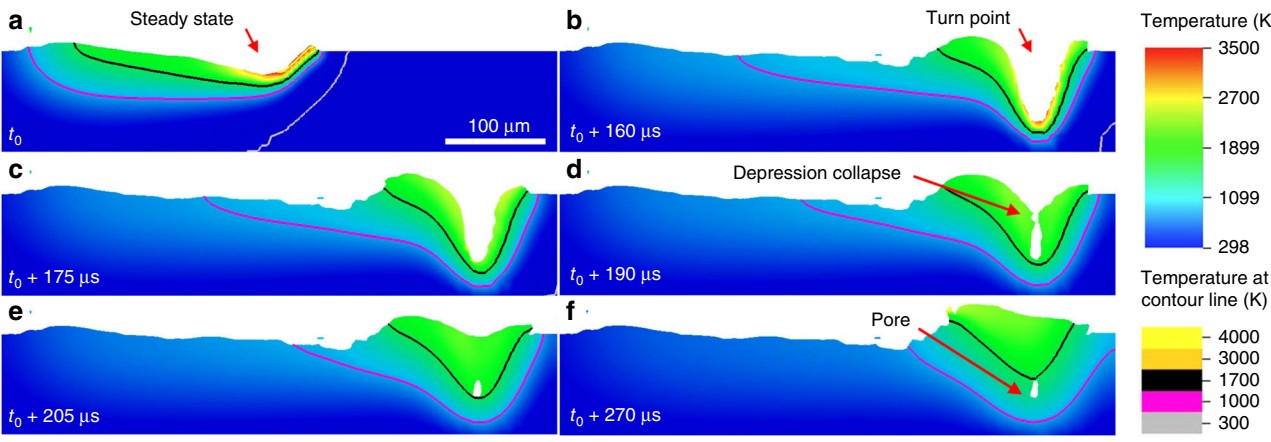

**Fig. 5** Cross-section of multi-physics simulation of turn point dynamics in SS316L performed using ALE3D. The laser power was 200 W, and the steady-state scan speed is 1500 mm s$^{-1}$ and reduced during the turn point as per the physical system. The black contour line represents the melt pool boundary

rapidly solidifying due to the extreme ($\sim10^6$ K s$^{-1}$) cooling rate at the pore location (Fig. 5e). The rapidly moving (0.3 m s$^{-1}$) solidification front thus traps gaseous material forming the pore (Fig. 5f) confirming our experimental observation. The simulations show that at the turn point the process enters a deep keyhole regime and large pore defects are generated as the process returns to steady state, agreeing with the experimental observations in Ti–6Al–4V. These complementary observations strongly suggest that this behavior is universal and occurs regardless of material in turn points during LPBF.

**Pore formation mitigation strategy**. From our experimental and modeling results, the increase in depth followed by a rapid collapse of the vapor depression is the mechanism that gives rise to the formation of pores during the laser turn point. This behavior is caused by a transient increase in energy density deposited by the laser and can therefore be mitigated by adjusting process parameters to keep energy density approximately constant. Turning the laser off at the turn point (a so-called sky writing method) is not a viable solution for pore mitigation as it has previously been shown to result in pore formation[16]. Incidentally, the formation of pores at the end of a written track when using the sky writing method can also be explained by the observations here describing the collapse of the vapor depression and the subsequent formation of pores. The vapor depression collapse when the laser is turned off likely exhibits behavior comparable to the conditions described due to the removal of laser power within the 6.5 μs measured for the Yb-fiber laser used in this study. In the laser off condition the transition from keyhole mode to zero laser input power would be even more abrupt than at the turn point. Given the physical limitations of the mirror-based laser scanning system, the only practical option available to stabilize the energy density during the turn point is to vary the laser power, which can be controlled at time scales on the order of 20 μs in modern Yb-fiber lasers. A scan strategy for mitigation of pore formation has the following mechanistic and physical requirements: (i) the vapor depression must not transition into the keyhole regime at the turn region, (ii) laser power must be controlled with no rapid oscillations, (iii) the power must be sufficient to maintain the melt pool during scanning of the hatch spacing, and (iv) the laser power must not increase rapidly when accelerating out of the turn point into the pre-heated region. A power profile strategy was devised to conform to these constraints and applied to LPBF of Ti–6Al–4V using a steady-state laser power of 100 W and a scan speed of 1000 mm s$^{-1}$ (Fig. 6a). Transmission X-ray imaging of the process shows that the

mitigation strategy results in a near uniform vapor depression depth during the entire scan pattern (see Supplementary Movie 2 and Supplementary Note 2 for in-process videos of the constant power and mitigation scan strategy cases). Most importantly, pores were not detected in the processed track using this scan strategy. When combined with contour and border hatch scan strategies which have been shown to reduce porosity[41] this power modulation scan strategy could further improve final component quality. This is particularly important in island scan sequences where the number of turn points per volume slice is increased significantly[42].

An analytical approach, utilizing normalized enthalpy[18,34,43], was used to investigate the outcomes of the mitigation strategy. Normalized enthalpy is a term commonly used in the welding literature[43,44] and recently has expanded to characterize LPBF conditions[18,34]. Previous studies have shown a linear dependence on the depth of molten material with normalized enthalpy under varied laser conditions. The normalized enthalpy $\left(\frac{\Delta H}{h_s}\right)$ is equal to $\frac{AP}{\pi\rho C T_m \sqrt{Dua^3}}$, where $\Delta H$ is the specific enthalpy, $h_s$ is the enthalpy at melting, $A$ is the absorptivity of the material (assumed to be 0.6 under all conditions), $P$ is the laser power, $\rho$ is the density (4.43 g cm$^{-3}$)[45], $C$ is the specific heat capacity (0.83 J g$^{-1}$ K$^{-1}$)[45], $T_m$ is the melting temperature (1923 K)[45], $D$ is thermal diffusivity of the molten material (0.086 cm$^2$ s$^{-1}$)[45], $u$ is the laser scan speed, and $a$ is the $\frac{1}{e}$ radius of the laser beam ($a = \sigma\sqrt{2}$). The normalized enthalpy approach was applied to the vapor depression depth as a function power data in Fig. 3a collected at varied laser scan speeds (see Supplementary Fig. 4). This reveals that the vapor depression depth is linear with normalized enthalpy and the relationship is identical for different laser scan speeds under these material processing conditions.

Normalized enthalpy as a function of laser position for the constant power and mitigation scan strategy are shown in Fig. 6b. During steady-state scanning at 1000 mm s$^{-1}$ and 100 W, the normalized enthalpy is approximately equal to 12.4. For the case of the turn point performed at constant power the normalized enthalpy is greater than the steady-state case in the region 500 μm before and after the turn point and reaches a maximum value of 63.2 at the turn. This increase in $\Delta H$ results in severe material overheating and the formation of a deep keyhole which ultimately leads to pore formation as the laser completes the turn. During the mitigated scan strategy, the normalized enthalpy is near constant, peaking at a value of 31.6 for approximately 100 μs at the turn point when the scan speed is reduced to 44 mm s$^{-1}$. The laser power was not reduced farther than 50 W in this region as

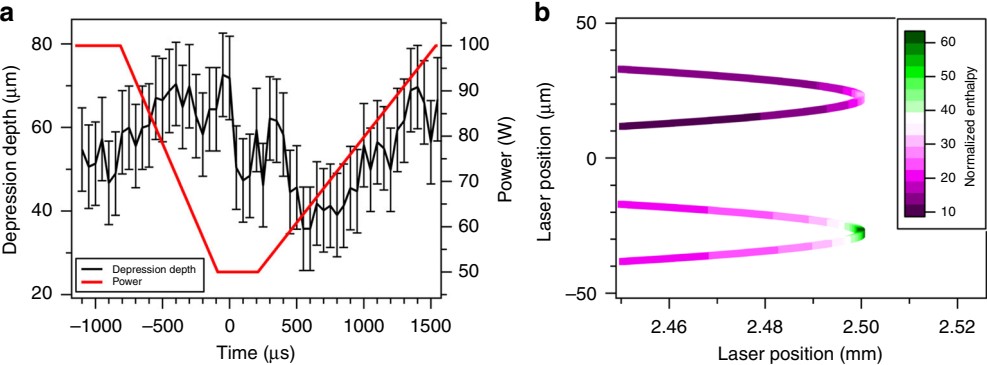

**Fig. 6** Mitigation of vapor depression depth change during LPBF of Ti–6Al–4V. **a** The black line (left axis) corresponds to vapor depression depth as a function of time during a 100 W peak power pore mitigation scan strategy. Error bars represent uncertainty in the distance between the base of the vapor depression and the surface caused by surface roughness. Also shown is the commanded laser power as a function of time used in the scan strategy (red line). Depression depth values were measured for the case of a bare plate experiment because depression depth measurements in bare plate were less uncertain than the powder case, but the same trend is observed in both cases. **b** Normalized enthalpy ($\frac{\Delta H}{h_s}$) represented by the magenta-green color scale as a function of laser position during the turn point for the full power and mitigated cases at 1000 mm s$^{-1}$ steady-state scan speed

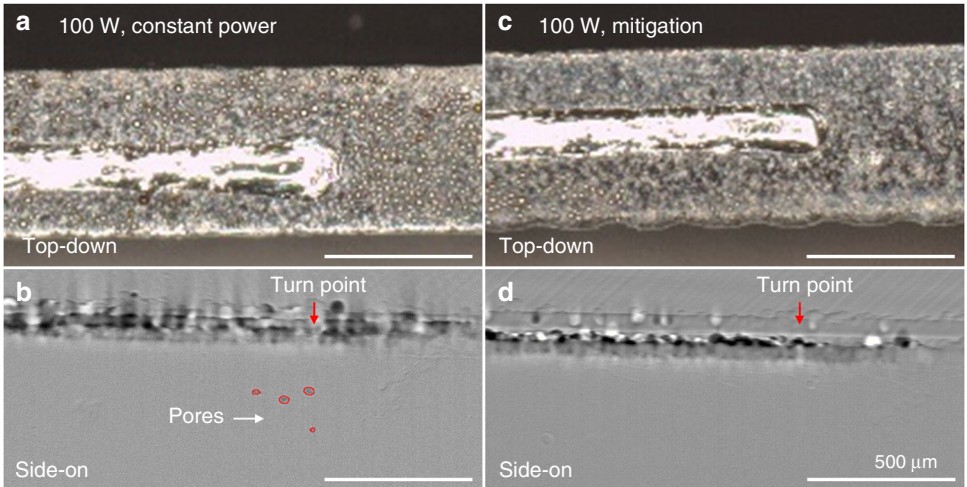

**Fig. 7** Quality of LPBF-AM tracks in Ti–6Al–4V. **a, b** Top-down optical image and side-on X-ray image of a LPBF-AM track in the turn point region produced using a constant power of 100 W. Pores formed during LPBF are highlighted in **b**. **c, d** Top-down optical image and side-on X-ray image of an LPBF track produced using the 100 W peak power pore mitigation scan strategy shown in Fig. 5b. No pores were detected in **d**. All scale bars are 500 µm

this approaches the cut-off power of the laser resulting in complete collapse of the depression and possible underheating. This short increase however did not result in a transition to a deep keyhole vapor depression likely due to thermal lag required to induce changes in overall melt pool behavior under these conditions. Regarding the effect of preheating, it is observed that under steady-state conditions the depression depth for all scan speeds probed is approximately 250 µm when the normalized enthalpy value equals 31.6 (see Supplementary Fig. 4). Comparing this behavior to the turn point case we observe that a brief increase to this level of normalized enthalpy for 100 µs during the turn does not cause the formation of a deep keyhole vapor depression and a depression depth of only 70 µm was realized. The analytical normalized enthalpy analysis reveals that for successful implementation of a defect mitigation scan strategy the normalized enthalpy should be kept near constant and below the transition point into the keyhole regime.

**Quality of tracks produced by pore mitigation strategy**. A pore mitigation strategy is not viable if the resulting track is of low quality (e.g., discontinuous). Figure 7 shows top-down optical and side-on transmission X-ray images of LPBF-AM turn point tracks produced on Ti–6Al–4V with a single 60 µm powder layer produced using constant power (100 W) and the mitigation strategy, respectively (for further examples see Supplementary Fig. 5). The track produced using constant power (Fig. 7a) exhibits a bulge at the turn point caused by overheating and expansion of the melt pool in this region with a track width 40% wider at the turn point than during the steady-state scan. The turn point using constant power also contains pores at depths up to approximately 250 µm beneath the substrate surface (Fig. 7b). For the case of the track produced using the mitigation strategy, Fig. 7c shows a clear improvement in track geometry. There is no longer a bulge at the end of the track caused by overheating. This reveals that not only is the formation of pores mitigated using this scan strategy (Fig. 7d), the quality of the resulting track with respect to the geometry and minimum track resolution is also improved. The mitigation strategy not only improves quality of material fabricated by LPBF-AM, but also is simple to implement, because it only requires linear ramping of the laser power over a few hundred microseconds. This sort of power adjustment can be implemented with the hardware available in most commercial LPBF machines by constructing power maps with the 3D slicer

software that converts the component geometry defined by a CAD file into machine instructions[46].

## Discussion

In summary, we have uncovered the mechanism of pore formation during laser turn points, a critical defect mode in serpentine scan-based LPBF. The pore formation process is observed experimentally in Ti–6Al–4V and via multi-physics modeling of SS316L, revealing the general nature of the mechanism. Pores form at laser turn points due to the emergence and subsequent collapse of a deep keyhole depression caused by the deceleration and acceleration of the galvanometer-based scanning mirrors during the turn which results in dramatic variations in the local normalized enthalpy at the material surface. As the laser accelerates away from the turn point, the keyhole depression collapses and molten metal fills the void, trapping gaseous argon which ultimately forms a pore as the material solidifies. This understanding based on in situ X-ray imaging and multi-physics modeling was harnessed to devise a pore mitigation strategy based on laser power modulation and implemented under typical Ti–6Al–4V build conditions. The mitigation strategy effectively prevents pore formation at the laser turn point by removing the rapid variation in depression depth inherent in the unmitigated case and improves the geometric tolerance of fabricated tracks by avoiding overheating. Conceptually similar strategies should be applicable to any abrupt laser on/off points during LPBF. The successful mitigation strategy presented here illustrates the potential of in situ X-ray measurements coupled with high fidelity modeling for driving process improvements and paves the way to increasing the quality of LPBF-built components.

## Methods

**LPBF system and processing conditions**. LPBF was performed using a laboratory-scale test bed described and characterized in detail elsewhere[22]. The LPBF system utilized a 1070 nm, continuous wave (CW) Yb-fiber laser (500 W maximum power, YLR-500-WC-Y14, IPG Photonics) coupled to a galvanometer scanning mirror system (Nutfield Technology, 3XB 3-Axis Scan Head) for processing. The laser was focused to a spot size of approximately 50 μm in diameter ($D4\sigma$) for all experiments and passed through an anti-reflective coated laser entry window into the vacuum chamber, normal to the sample surface. The optical working distance between the scanning mirrors and the sample surface was approximately 380 mm, and the laser beam Rayleigh range was 1.8 mm. The vacuum chamber containing the sample was evacuated to $5 \times 10^{-2}$ Torr prior to being filled with 730 Torr argon inert gas environment for processing. Argon was constantly flowed through the vacuum chamber during experiments at 500 SCCM. During processing, the laser was scanned using various laser power and scan speed conditions onto a region of a Ti–6Al–4V substrate (TMS Titanium, Poway, CA, USA). Each substrate was approximately 500 μm thick in the X-ray probe direction and 10 mm in depth. Experiments were performed with and without a 60 ± 20-μm-thick layer of Ti–6Al–4V powder (30 ± 10 μm powder diameter; Additive Metal Alloys, Maumee, OH, USA) on the surface. The Ti–6Al–4V substrate was sandwiched between two 1-mm-thick glassy carbon sheets which provided a trench to contain Ti–6Al–4V powder on the substrate surface.

Laser turn-around scanning conditions were programmed using the Waverunner scan control software and Pipeline-2 scan controller (Nutfield Technology) which compiled the required instruction routine for the galvanometer scanning mirrors and laser power interface. Two parallel, 2.5 mm long tracks were compiled in the software with a hatch spacing of 50 μm. The laser was programmed to irradiate this geometry based on internal triggering from the scanning mirror position. The geometry treated the hatching shift at the end of the track as an additional 50 μm long track and unless stated, the laser remained at full power during the turn. The mitigated scan strategy was implemented using a custom field-programmable gate array (FPGA)-based laser interface module (USB-7856; National Instruments). The galvanometer scanning mirrors were controlled by the Pipeline-2 controller for all cases, with mitigation achieved by disabling the Pipeline-2 controller laser power interface and initializing the FPGA module to control the laser power via an analog voltage signal. The FPGA module controlled the laser power as a function of time using a lookup table. Scanning mirror position was sampled at a rate of 1 MHz using an FPGA module, and the analog output converted to position via a calibration routine.

**Imaging and data processing**. In situ X-ray imaging was performed at SSRL beam line 2-2. The white-beam X-ray spectrum generated by the 1.25 T bend magnet was

utilized for the experiments (X-ray critical energy 7.4 keV). The beam was aligned coincident with the Ti–6Al–4V substrate surface in the center of the vacuum chamber and the laser aligned to scan through the X-ray imaging system field of view during processing. Transmission X-ray images of the LPBF process were captured using a scintillator-based optical system. The imaging system comprised an X-ray shutter (Uniblitz), 100-μm-thick YAG:Ce scintillator crystal (Crytur), Ag-coated turning mirror (Thorlabs), 10× long working distance infinity corrected objective lens (0.28 NA; Mitutoyo), tube lens (Thorlabs), and FASTCAM SA-X2 1080 K high-speed camera (Photron). This imaging assembly yields an effective pixel size of 2 μm per pixel for all X-ray images. Images were captured with a field of view of 1024 × 672 pixels at 20 kHz and an exposure time of 25 μs. The X-ray shutter was placed in front of the scintillator and actuated to the open position approximately 50 ms before the laser entered the field of view and then closed after a total time of 150 ms to protect the detector system from damage by the X-ray beam. Synchronization of the laser and imaging system was realized using a custom FPGA-based timing circuit.

X-ray images were analyzed using ImageJ[47] and Mathematica (Version 11.1.1)[48] software packages. Time difference X-ray images were produced through division of the uncorrected time resolved image ($A_t$) by the initial, pristine substrate image ($A_0$) (image $= \frac{\ln(A_t)}{\ln(A_0)}$). This routine provided an image where darker regions reveal a decrease in X-ray absorption (or material) and lighter regions reveal an increase in X-ray absorption (or material). A custom script in Mathematica utilized the built-in Binarize contrast threshold method to identify and characterize pores in the processed Ti–6Al–4V substrate. Optical images of processed tracks were captured ex situ using a Keyence VR-3000 wide-area 3D measurement system and analyzed using the Keyence VR-3000 G2 and ImageJ software packages.

**Multi-physics simulation**. Simulations were performed using the ALE3D multi-physics software tool which utilizes arbitrary-Lagrangian-Eulerian techniques[39,49]. To simulate the thermal response and melt flow in the material, the heat conduction equation was coupled with the Navier–Stokes equations via operator splitting[17]. Material parameters for stainless steel (316L) were used for the simulation[17]. The substrate was modeled as a flat, bare plate surface with a boundary of 600 × 300 × 100 μm. Powder was not included as it significantly increases the physical complexity of the simulation and experimental results showed no significant change in the trend in pore formation between bare plate and powder. The simulation was performed using a high-resolution mesh to enable a feature detection limit of 3 μm. A series of simulations was performed as a function of laser power during the turn point. Under each condition, a turn point track was simulated where the laser power reduced from a steady-state laser power of 200 W to a constant value during the turn point. At each 1 μs simulation time step the full thermal and laser reflection profile in the material was recorded. The maximum melt depth as a function of time was determined as a function of turn point laser power. All simulations were performed using a steady-state scan speed of 1500 mm s$^{-1}$, and assuming a constant absorptivity of 0.25. A scan speed of 1500 mm s$^{-1}$ was required due to the exhaustive computational requirements of multi-physics simulation. The laser followed a scan geometry and turn point scan speed informed by measurements of the galvanometer scan mirror response, which comprised of a straight line stretch ending in a turn point. The speed of the laser through the turn was reduced to 400 mm s$^{-1}$ to ensure the simulation stayed within bounds (see Supplementary Fig. 3).

## Data availability

The data that support the findings of this study are available from the corresponding author on reasonable request.

## Code availability

The ALE3D software routine and custom Mathematica script are not publicly available. All data generated using this code are available from the corresponding author on reasonable request.

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

## Acknowledgements

This material is based upon work supported by the U.S. Department of Energy's Office of Energy Efficiency and Renewable Energy (EERE) under the Advanced Manufacturing Office, CPA agreements 32035, 32037, and 32038. Lawrence Livermore National Laboratory (LLNL) is operated by Lawrence Livermore National Security, LLC, for the U. S. Department of Energy, National Nuclear Security Administration under Contract DE-AC52-07NA27344. Use of the Stanford Synchrotron Radiation Lightsource, SLAC National Accelerator Laboratory, is supported by the U.S. Department of Energy, Office of Science, Office of Basic Energy Sciences under Contract No. DE-AC02-76SF00515. The authors acknowledge experimental assistance from Doug Van Campen, Ron Marks, Tim J. Dunn, and Matthew Latimer, sample preparation by the LLNL Precision Machine Shop, graphics assistance by Veronica Chen at LLNL, and helpful discussions with Matthew Kramer, Peter Collins, Ryan Ott, Jianchao Ye, and Wayne King.

## Author contributions

A.A.M, N.P.C. and S.A.K wrote the article with contributions from all authors. A.A.M., N.P. C., J.W., P.J.D., A.Y.F., V.T. and A.M.K. performed the in situ X-ray imaging experiments. J.W. performed the optical imaging measurements. G.M.G. and A.A.M. contributed to laser control electronics design. S.A.K. performed the multi-physics simulations. M.J.M. and T.v.B. supervised work at LLNL. M.F.T., J.N.W., C.J.T. and K.H.S supervised work at SSRL.

## Additional information

**Competing interests:** The authors declare no competing interests.

