## [Peer Review File · Nature Communications]

Reviewers' comments:

Reviewer #1 (Remarks to the Author):

A problem of the turn point mentioned in this paper is one of critical issues for practical use of AM. However, there is a trend of steady state scan speed in digital technology. Therefore, the point focused in this paper is very interesting from the viewpoint of digital manufacturing.

The problem derived from rapid increase and decrease of laser energy density as change of scan speed around the turn point. So modulation of the laser power is effective.

Regarding X ray observation, series of images captured at 20 kHz are new data in comparison with the conventional X-ray observations from 1 kHz to 100 Hz. The multi-physics simulation demonstrates the clear relationship between depression collapse and pore. These results of this paper are simple and reasonable.

I do see the need for some clarifications and smaller corrections, however, and I hope that you can share my arguments below.

1. Concerning 50 μm in diameter at line 380, the figure of laser beam profile used in this experiment must be attached as function of defocused distance for wide range of readers.
2. Concerning the physics of laser –metal interaction at line 21, What exactly do you mean by the physics? It is difficult to understand the physics by reading this paper.
3. Results of X-ray observation is recognized at first glance. They should be written in detail for wide range of readers. Judging from images in Fig.1 (d) to (f) at line 83, there are the following possible causes of the pore formation. The cause is that molten metal cannot track the rapid growth and shrinkage of vapor depression at tuning point. This led to the depression collapse. Why not mention the mechanism of pores formation at section of Pores formed at the laser turn point? Moreover, information inside pore is needed. Vacuum or argon gas filled?
4. Concerning that the vapor depression depth increase linearly with laser power at line 141, let me know why linear increase with not energy(J) but power (W: J/s). The question remains unsettled in laser materials processing. You should write down the reason here.
5. Concerning absorption in multi-physics simulation, there is a question.

Fresnel equations describe the reflection, absorption and transmission of laser beam. Is the laser-metal absorption in the simulation taken into account of Fresnel equations? You should describe the value used as the absorption between line 228 and 237 in order to understand for wide range of reader.

6. Concerning vapor in multi-physics simulation, there is a question.

The depression is made by the balance between surface tension of molten pool and recoil pressure of metal vapor. You should describe the value of metal vapor between line 228 and 237 in order to understand for wide range of reader. Moreover, Is the simulation taken into account of readhesion of the metal vapor? If not, is the simulation under energy conservation? You should describe your opinion between line 228 and 237 in order to understand for wide range of reader.

7. Concerning laser powder bed in multi-physics simulation, there is a question.

A laser beam travels in space between powders in the bed. Is the simulation taken into account of travel of the laser beam between the powders? This point is important, because, if not, the simulation is almost the same as the conventional simulation of laser welding. You should describe effects of travel of the laser beam between the powders for wide range of reader.

Reviewer #2 (Remarks to the Author):

Dear authors, thanks for this great contribution. Your research proves clearly the mechanisms of pore formation near laser turn points and provides with an elegant (machine-agnostic) solution to mitigate the formation of such porosity. This research is certainly very useful to the additive community.

Two main comments:

1) It is probably worth to acknowledge in the manuscript how the issue of porosity near turn points is commonly addressed. As per current industry practice each slice is typically scanned by a number of i) borders (laser scans that outline incrementally the area of each slice), ii) contours (laser scans that are offset further from the border scans) and iii) volume hatches. Border (and contour) hatches are always applied (even in the case of complex scan strategies like chessboard) to i) improve surface finish and ii) eliminate porosity near edges. For info see image attached.

2) Your research seems to show that porosity only forms after to a laser turn point (we read statements like "No pores were formed in the turn point region prior to the laser turn in these experiments" and "the steady state vapor depression depth during the post-turn scan is higher than the pre-turn scan due to thermal lag associated with pre-heating of the material"). This is somehow in contrast to the data presented in Figure 4 where the depression depth just before (-200us) and after the turn point (+500us) is comparable. For example, in Figure 4 red circles mark the events where pore formation has occurred: these points correspond to depression depths that are certainly comparable to those observed prior the turn point. Am I missing something here?

Minor comments:

1) For sake of clarity I believe that Figure 1 should be moved to the result chapter. In the present manuscript Figure 1 is part of the introduction. It is worth to explicit in the caption that images d) to f) are transmission X-ray images?

2) Figure 3: the depression depths obtained at laser powder 50, 75 and 100W are probably a result of conduction-mode melting (and not key-hole melting) as at these power levels no (or limited) vaporisation of the metal surface should be expected. Is it worth to double check your dataset? If obtained under conduction-mode melting, it could be argued that these points should be excluded from the graph as the melt pool geometry would be governed by different physics.

Thanks,

Marco Simonelli

Decision on manuscript NCOMMS-18-34687

Title: Dynamics of pore formation during laser powder bed fusion additive manufacturing

Reviewer 1

A problem of the turn point mentioned in this paper is one of critical issues for practical use of AM. However, there is a trend of steady state scan speed in digital technology. Therefore, the point focused in this paper is very interesting from the viewpoint of digital manufacturing. The problem derived from rapid increase and decrease of laser energy density as change of scan speed around the turn point. So modulation of the laser power is effective. Regarding X ray observation, series of images captured at 20 kHz are new data in comparison with the conventional X-ray observations from 1 kHz to 100 Hz. The multi-physics simulation demonstrates the clear relationship between depression collapse and pore. These results of this paper are simple and reasonable.

Comments:

I do see the need for some clarifications and smaller corrections, however, and I hope that you can share my arguments below.

R1.1. Concerning 50 μm in diameter at line 380, the figure of laser beam profile used in this experiment must be attached as function of defocused distance for wide range of readers.

We understand the concern for the reviewer regarding the laser beam profile as a function of defocused distance and how this may potentially affect our results. There is certainly concern over the large scan areas used in industrial laser powder bed fusion systems that the focal distance can change from the center of the build area to the periphery resulting in beam profile changes. In our case the track length is only 2.5 mm and the Rayleigh range (defocus distance where the area of the beam cross section is doubled) of the beam is 1.8 mm. The working distance between the scanning mirrors and the metal surface is on the order of 380 mm. Therefore, the change in focal length between the start and end of track is only 8 μm which is difficult to measure experimentally and has negligible effect on the beam profile. As such we believe that the defocused distance measurement does not add any value to the manuscript. In any case, additional information has been added to the manuscript to describe the Rayleigh range and optical working distance of the laser scanning system for readers to refer to (Page 14, Paragraph 2).

R1.2. Concerning the physics of laser –metal interaction at line 21, What exactly do you mean by the physics? It is difficult to understand the physics by reading this paper.

We use the term physics in the most general sense, that is, the interaction of forces or fields with matter. In our case we are referring to the motion of the unstable fluid-fluid interface under the action of vapor recoil and surface tension gradients. As we point out, our results reveal the timing of the pore collapse in laser keyhole processing during changes in scan behavior. Furthermore, we show that the salient physics is well-described in our finite element model, despite some of the approximations used. This is important because without experimental verification, the applicability of approximate models would remain unknown.

R1.3. Results of X-ray observation is recognized at first glance. They should be written in detail for wide range of readers. Judging from images in Fig.1 (d) to (f) at line 83, there are the following possible causes of the pore formation. The cause is that molten metal cannot track the rapid growth and shrinkage of vapor depression at tuning point. This led to the depression collapse. Why not mention the mechanism of pores formation at section of Pores formed at the laser turn point? Moreover, information inside pore is needed. Vacuum or argon gas filled?

We assume the reviewer means not recognized at first glance. We have therefore added a short statement in the results section to clarify how the data for Fig. 2 was produced (Page 4, Paragraph 1). More information regarding X-ray image formation and analysis is stated in the Methods section of the manuscript.

We are a little uncertain as to the reviewer's comment regarding "Why not mention the mechanism of pores formation at section of Pores formed at the laser turn point." In lines 361-364 of the manuscript we state, "Pores form at laser turn points due to the emergence and subsequent collapse of a deep keyhole depression caused by the deceleration and acceleration of the galvanometer-based scanning mirrors during the turn which results in dramatic variations in the local normalized enthalpy at the material surface."

Regarding the nature of the pores, vacuum or argon filled, we turn to previous laser welding studies. Based on these studies the vapor inside the keyhole is mostly metal vapor with a small portion of inert shielding gas (in our case argon) [1]. As the depression collapses this inert gas is trapped in the pore. These findings from the laser welding literature have been applied in the literature to describe LPBF by other groups (Ref. 14). We have added the additional reference from the welding literature (now Ref. 19) to our original statement at lines 51-52 of the manuscript "The keyhole depression is unstable and can collapse to trap inert shielding gas, such as argon, in pores within the substrate."

[1] Katayama, S., Seto, N., Kim, J.-D. & Matsunawa, A. Formation mechanism and reduction method of porosity in laser welding of stainless steel. Int. Congr. Appl. Lasers Electro-Opt. 1997, G83 (1997).

R1.4. Concerning that the vapor depression depth increase linearly with laser power at line 141, let me know why linear increase with not energy(J) but power (W: J/s). The question remains unsettled in laser materials processing. You should write down the reason here.

The nature of energy or power scaling in laser materials processing is indeed an interesting and evolving subject of debate. Our group has sought to clarify energy and material scaling relations for additive manufacturing (also applicable to laser welding), see for example Ref. 34 of the manuscript or [2]. In fact, we have found that the depth scales more accurately with normalized energy (enthalpy) density and not simply laser power, which allows scaling across different materials and laser parameters. However, for the same material and constant laser beam size and scan speed, melt pool depth is found to scale reasonably with laser power. A linear scaling of vapor depression depth with laser power is expected for conduction mode processing but is not necessarily expected for keyhole mode heat transport where strong vaporization and melt pool dynamics play important roles. In this keyhole regime, the melt rapidly displaces away from the laser beam under the effect of recoil momentum and Marangoni shear flow. Therefore, the absorbed laser energy not only leads to melting, but also to melt motion. For example, in the case of a SS316L melt pool with a velocity of 10 m/s, the kinetic energy density ($(\rho v^2)/2$), where v is the velocity of the melt) is less than 1 J/cm^3 , which is three orders of magnitude smaller than the melt enthalpy. Therefore, we argue that the absorbed laser energy is spent mainly to melt the solid even in the keyhole regime, which helps to explain the linear scaling behavior with power observed in both thermal conduction and keyhole regimes.

We have added further discussion regarding the linear scaling of the vapor depression to the manuscript (Page 6, Paragraph 1).

[2] King, W. E. et al. Laser powder bed fusion additive manufacturing of metals; physics, computational, and materials challenges. *Appl. Phys. Rev.* 2, 041304 (2015).

R1.5. Concerning absorption in multi-physics simulation, there is a question. Fresnel equations describe the reflection, absorption and transmission of laser beam. Is the laser-metal absorption in the simulation taken into account of Fresnel equations? You should describe the value used as the absorption between line 228 and 237 in order to understand for wide range of reader.

The laser metal interaction model does not use the Fresnel equations since it does not involve full laser ray tracing. The rays strike the surface from the Gaussian beam source in a direct line of sight and the energy deposited into the sample is determined by an effective absorption coefficient (0.25 for a bare surface). From Ref. 17, the bulk of the incident laser energy is deposited over the front inclined wall of the vapor depression which consists of a flat (inclined) liquid surface. The energy deposited into the front wall location dominates the melt pool response and melt pool depth via the recoil pressure. This is due to the exponential temperature dependence of the recoil physics involved [3] and because the highest temperature is achieved immediately below the laser at the point of incidence. Although the angle between the incident rays and the melt pool surface vary considerably, it can be shown that for unpolarized laser light (e.g. the Yb-fiber laser used in our study), the angular dependence due to Fresnel reflectivity is negligible for most metals up to $\sim 60^\circ$. This is not the case for purely P- or S-polarized light. We also note that the rays that are incident are only simulated and do not undergo reflections. This helps to economize on the large simulation expense. These details have been added to the manuscript (Page 9, Paragraph 1).

We note, the model is described in detail our groups article "Laser powder-bed fusion additive manufacturing: Physics of complex melt flow and formation mechanisms of pores, spatter, and denudation zones" (Ref. 17 of the manuscript). Section 2.1 and Figs. 3 and 4 of Reference 17 completely describe the point of incidence of the rays. The recoil physics are described by Anisimov's model as detailed in reviewer response R1.6.

[3] Anisimov, S. I. & Khokhlov, V. A. *Instabilities in Laser-Matter Interaction*. (CRC Press, 1995)

R1.6. Concerning vapor in multi-physics simulation, there is a question. The depression is made by the balance between surface tension of molten pool and recoil pressure of metal vapor. You should describe the value of metal vapor between line 228 and 237 in order to understand for wide range of reader. Moreover, Is the simulation taken into account of readhesion of the metal vapor? If not, is the simulation under energy conservation? You should describe your opinion between line 228 and 237 in order to understand for wide range of reader.

The peak value of the evaporative flux occurs directly below the laser spot on the front incline of the depression and is approximately $1000 \text{ mol/m}^2\text{s}$. The model solves the Navier-Stokes equations coupled with the heat diffusion equation in an energy conserving scheme, while accounting for the vapor recoil pressure as well as the evaporative cooling as boundary conditions through Anisimov's model [3].

According to Anisimov's model, approximately 18% of the metal vapor condenses back to the surface due to large angle scattering collisions in the vicinity of the liquid and hence reduces the evaporative cooling effect. The net material evaporation flux is $J_v = \frac{0.82AP(T)}{\sqrt{2} \pi MRT}$ and is consistent with the recoil pressure, $P(T)$, derivation. A is a sticking coefficient, which is close to unity for metals, M is the molar mass, R the gas constant and T the surface temperature. More advanced details can be found in Section 2.2 of Reference 17 for readers requiring further background on the simulation.

As per the reviewer's request these details have been added to the manuscript (Page 9, Paragraphs 1 and 2) and further descriptions are now available in the Supplementary Information.

[3] Anisimov, S. I. & Khokhlov, V. A. *Instabilities in Laser-Matter Interaction*. CRC Press, Boca Raton, Florida, (1995).

R1.7. Concerning laser powder bed in multi-physics simulation, there is a question. A laser beam travels in space between powders in the bed. Is the simulation taken into account of travel of the laser beam between the powders? This point is important, because, if not, the simulation is almost the same as the conventional simulation of laser welding. You should describe effects of travel of the laser beam between the powders for wide range of reader.

As mentioned in the manuscript (lines 432-434) "Powder was not included as it significantly increases the physical complexity of the simulation and experimental results showed no significant change in the trend in pore formation between bare plate and powder."

If powder is used in the model, the simulation does not resolve laser beam travel in space between powder particles in the bed. Due to rapid powder melting ($\sim \mu\text{s}$ time scale) and powder denudation (Ref. 24) it turns out that the laser mostly couples to the liquid metal surface during steady-state scanning conditions and not directly to the powder. This is described in detail in Fig. 3 of Reference 17.

Reviewer 2

Dear authors, thanks for this great contribution. Your research proves clearly the mechanisms of pore formation near laser turn points and provides with an elegant (machine-agnostic) solution to mitigate the formation of such porosity. This research is certainly very useful to the additive community.

Comments:

Two main comments:

R2.1. It is probably worth to acknowledge in the manuscript how the issue of porosity near turn points is commonly addressed. As per current industry practice each slice is typically scanned by a number of i) borders (laser scans that outline incrementally the area of each slice), ii) contours (laser scans that are offset further from the border scans) and iii) volume hatches. Border (and contour) hatches are always applied (even in the case of complex scan strategies like chessboard) to i) improve surface finish and ii) eliminate porosity near edges. For info see image attached.

We thank the reviewer for pointing out this important build parameter to reduce porosity. There has certainly been a great deal of research in identifying how scan parameters affect porosity, particularly at the edge of a part. Examples include research performed by the National Institute of Standards and Technology [4] and the University of Waterloo [5]. We were unable to find examples of where borders and contours eliminate pores completely. Perhaps future research could use contour scan strategies with the power mitigation strategy for further reduction of porosity in full builds. We predict that the power mitigation strategy will be particularly useful in “island” scan strategies such as the chessboard pattern mentioned by the reviewer which includes significantly more turn points as the inner portion of the volume slice is broken up into many small scan islands.

We have included a statement and references in the manuscript referring to scan strategies leading to porosity reduction (Page 11, Paragraph 1).

[4] Yeung, H., Lane, B., Fox, J., Kim, F., Heigel, J. & Neira, J. Continuous laser scan strategy for faster build speeds in laser powder bed fusion system. in Proceedings of the 28th Annual International Solid Freeform Fabrication Symposium 1423–1431 (Laboratory for Freeform Fabrication and University of Texas at Austin, 2017)

[5] Ertay, D. S., Ma, H. & Vlasea M. Correlative beam path and pore defect space analysis for modulated powder bed laser fusion process. in Proceedings of the 29th Annual International Solid Freeform Fabrication Symposium 274–284 (Laboratory for Freeform Fabrication and University of Texas at Austin, 2018)

R2.2. Your research seems to show that porosity only forms after to a laser turn point (we read statements like "No pores were formed in the turn point region prior to the laser turn in these experiments" and "the steady state vapor depression depth during the post-turn scan is higher than the pre-turn scan due to thermal lag associated with pre-heating of the material"). This is somehow in contrast to the data presented in Figure 4 where the depression depth just before (-200 μ s) and after the turn point (+500 μ s) is comparable. For example, in Figure 4 red circles mark the events where pore formation has occurred: these points correspond to depression depths that are certainly comparable to those observed prior the turn point. Am I missing something here?

As stated in the caption of Figure 4 "time zero corresponds to the mid-point of the turn". The two times stated by the reviewer, -200 and +500 μ s, are actually during the turn, not in steady state. Steady state scanning can be compared between depression depths at -1000 and +1300 μ s. Comparing these two time points reveals that the post-scan vapor depression is slightly deeper than the pre-scan. As noted in lines 184-186 of the manuscript the change in scan depth can be directly observed in Fig. 1a and c.

To clarify the different scan regimes and when the turn occurs, we have added a description to the caption of Figure 4.

Minor comments:

R2.3. For sake of clarity I believe that Figure 1 should be moved to the result chapter. In the present manuscript Figure 1 is part of the introduction. It is worth to explicit in the caption that images d) to f) are transmission X-ray images?

As per the reviewer's suggestion Figure 1 has been moved to the results section. We agree this is more appropriate as we call the figure during that section of the manuscript.

The reviewer perhaps missed the portion of the caption (line 87) which states "(d-f) Time difference ($t-t_0$), transmission X-ray images of a turn point region in Ti-6Al-4V performed at a laser power of 200 W and scan speed of 1000 mm s⁻¹."

R2.4. Figure 3: the depression depths obtained at laser powder 50, 75 and 100W are probably a result of conduction-mode melting (and not key-hole melting) as at these power levels no (or limited) vaporisation of the metal surface should be expected. Is it worth to double check your dataset? If obtained under conduction-mode melting, it could be argued that these points should be excluded from the graph as the melt pool geometry would be governed by different physics.

Some care needs to be taken with comparing conduction-mode and keyhole regime melting. During keyhole regime melting the laser drills deep into the substrate as a result of strong vaporization of metal [1]. The fact that we directly observe the emergence of a vapor depression at all laser powers including 50 W means that we are at the very least in the keyhole transition regime at these powers. It is certainly interesting that the relationship is linear over the wide power range and we have certainly confirmed the dataset was accurately measured. As detailed in our response to reviewer comment R1.4 "The absorbed laser energy is spent mainly to melt the solid even in the keyhole regime, which helps to explain the linear scaling behavior with power observed in both thermal conduction and keyhole regimes."

REVIEWERS' COMMENTS:

Reviewer #1 (Remarks to the Author):

This revised paper accept all suggested corrections.
Therefore, I recommend that it be accepted for publication.

Reviewer #2 (Remarks to the Author):

Dear authors,

thanks for your comprehensive reply. Well done - looking forward to seeing your strategy implemented in commercial machines. Another step towards a better understanding of laser powder bed fusion.

Marco Simonelli